# Identification and characterization of recent retrovirus in *Rhinolophus ferrumequinum* bats

Hee Chun Chung,[1] Sung Jae Kim,[2] Su Jin Hwang,[1,3] Young Shin Jeon,[4] Min Sik Song,[5] Si Hwan Ko,[1] Jasper Lee,[6] Yoona Choi,[1,3] Chul Un Chung,[4] Jae Myun Lee[1,3]

**ABSTRACT** An investigation into retrovirus was conducted in six species of bats (*Myotis aurascens*, *Myotis petax*, *Myotis macrodactylus*, *Miniopterus fuliginosus*, *Rhinolophus ferrumequinum*, and *Pipistrellus abramus*) inhabiting South Korea. Exogenous retroviruses (XRVs) were detected in the tissue samples of *R. ferrumequinum* individuals by PCR assay. Proviruses were identified in all tissue samples through viral quantification using a digital PCR assay per organ (lung, intestine, heart, brain, wing, kidney, and liver), with viral loads varying greatly between each organ. In phylogenetic analysis based on the whole genome, the Korean bat retroviruses and the *R. ferrumequinum* retrovirus (RfRV) strain formed a new clade distinct from the *Gammaretrovirus* clade. The phylogenetic results determined these viruses to be RfRV-like viruses. In the Simplot comparison, Korean RfRV-like viruses exhibited relatively strong fluctuated patterns in the latter part of the envelope gene area compared to other gene areas. Several point mutations within this region (6,878–7,774 bp) of these viruses were observed compared to the RfRV sequence. One Korean RfRV-like virus (named Y4b strain) was successfully recovered in the Raw 264.7 cell line, and virus particles replicated in the cells were confirmed by transmission electron microscopy. RfRVs (or RfRV-like viruses) have been spreading since their first discovery in 2012, and the Korean RfRV-like viruses were assumed to be XRVs that evolved from RfRV.

**IMPORTANCE** *R. ferrumequinum* retrovirus (RfRV)-like viruses were identified in greater horseshoe bats in South Korea. These RfRV-like viruses were considered exogenous retroviruses (XRVs) that emerged from RfRV. Varying amounts of provirus detected in different organs suggest ongoing viral activity, replication, and *de novo* integration in certain organs. Additionally, the successful recovery of the virus in the Raw 264.7 cell line provides strong evidence supporting their status as XRVs. These viruses have now been identified in South Korea and, more recently, in Kenya since RfRV was discovered in China in 2012, indicating that RfRVs (or RfRV-like viruses) have spread worldwide.

**KEYWORDS** bat, retrovirus, detection, *Rhinolophus ferrumequinum* retrovirus, phylogenetic analysis

Retroviruses (family *Retroviridae*), a diverse group of RNA viruses, exhibit a remarkable ability to infect a wide range of species, including humans, animals, and plants (1). Their unique replication strategy involves reverse transcription of their RNA genome into DNA, which is then integrated into the host cell's genome (2). Endogenous retroviruses (ERVs) are ubiquitously present across diverse species (3) and constitute a fascinating aspect of retrovirus biology. This involves the integration of viral genetic material into the host genome over evolutionary time (4). These integrated sequences, remnants of past infections, play essential roles in shaping host genomes and have been implicated in various physiological processes (5). On the other hand, exogenous retroviruses (XRVs)

Address correspondence to Hee Chun Chung, heeskyi@yuhs.ac, Chul Un Chung, batman424@naver.com, or Jae Myun Lee, jaemyun@yuhs.ac.

Hee Chun Chung and Sung Jae Kim contributed equally to this article. The order was decided after mutual agreement.

The authors declare no conflict of interest.

See the funding table on p. 16.

are actively infectious, utilizing their reverse transcription machinery to replicate and propagate within host cells, often causing diseases or contributing to genetic diversity in susceptible populations. Recently, Koala retrovirus (KoRV) is distinctive in its dual nature, manifesting as both an exogenous and an actively endogenizing *Gammaretrovirus* within the koala population (6). Retroviruses are classified into several genera, each characterized by distinct genetic and structural features (5, 7, 8). Notable genera include *Alpharetrovirus*, *Betaretrovirus*, *Gammaretrovirus*, *Deltaretrovirus*, *Epsilonretrovirus*, *Spumavirus*, and *Lentivirus* (8, 9). The exploration of retrovirus genera provides a foundation for comprehending the intricate dynamics of viral evolution, transmission, and pathogenesis (10). *Alpharetroviruses* are associated with avian hosts (11), while *Lentiviruses*, such as HIV, are known for their ability to cause persistent infections in mammals (12, 13). *Gammaretroviruses* have been implicated in various diseases in animals, with examples such as murine leukemia viruses (14–16). Their capacity for insertional mutagenesis and oncogenesis underscores their significance in both basic virology research and as potential agents in the development of gene therapy vectors (17, 18). Among various animals, bats are notably considered a treasure trove of viruses (19, 20), harboring a diverse array of viral species. A novel retrovirus named *Rhinolophus ferrumequinum* retrovirus (RfRV) was reported from bats by Cui et al. (21) in 2011, providing significant genetic insights. Nevertheless, subsequent studies on the nature of retroviruses in bats are still limited, indicating a lack of current understanding, especially, concerning RfRV, which is the first discovered bat retrovirus. Notably, there have been reports of the discovery of retrovirus sequences in bats between 2011 and 2021, but there is no report of isolation or proliferation of the bat retrovirus (22, 23). In this study, we explored retroviruses in various bat species in South Korea, elucidating the characteristics of bat retroviruses through bioinformatics and molecular analyses.

## RESULTS

### Detection of bat retrovirus

Bat retroviruses were identified in all *R. ferrumequinum* individuals subjected to this study, while no retrovirus was detected in other bat species (Table 1; Fig. 1). Based on these results, bat viruses were strongly presumed to be a type of RfRV. Because these viruses were detected by PCR in DNAs extracted from tissue samples, it could not be determined whether they are ERV or XRV types. Unfortunately, clean serum or plasma samples, useful for determining the presence of XRVs, were not obtained in this study. This limitation arose from the complete coagulation of blood in the carcasses, rendering it unsuitable for PCR assays.

To find evidence for determining the type (ERV or XRV) of these viruses, viral loads in several organs were investigated using digital PCR (dPCR). Tissue samples (lung, intestine, heart, brain, wing, kidney, and liver) from six individuals (Y3, Y4, Y6, Y7, Y8, and Y14) were subjected to the dPCR assay, revealing the presence of proviruses in all tissue samples with considerable variation in viral loads. These results indicated that this virus was actively replicating within the host implying that Korean-RfRV viruses are XRVs. Notably, the highest total proviral copy numbers were found in the intestine samples, implying that a large number of viruses are excreted via feces, likely serving as fomites of transmission ($P < 0.05$; Fig. 2).

### Likelihood mapping to establish the best fit data set for phylogenetic analysis

In partitioned triangular graphs generated by likelihood mapping analysis, the three corner regions signify well-resolved phylogeny (informative), and the central triangle represents a star-like evolutionary pattern (uninformative). The three rectangles represent a partly resolved phylogeny (partly informative). In general, a good data set should show a high percentage of the sum of the three corner regions and a low percentage of the central region (24). In the likelihood mapping on all retroviral

**TABLE 1** Sampling information and detection for bat retrovirus

| Sample number | | Collection date (yr-mo-day) | Sex | Species | Province (city) | Sample type | Bat retrovirus | GenBank accession number. |
|---|---|---|---|---|---|---|---|---|
| Y1 | a | 2022-09-13 | Female | *Myotis aurascens* | Chungnam (Cheonan) | Lung | − | |
| | b | | | | | Intestine | − | |
| Y2 | a | 2022-09-13 | Female | *M. aurascens* | Chungnam (Cheonan) | Lung | − | |
| | b | | | | | Intestine | − | |
| Y3 | a | 2022-08-17 | Male | *R. ferrumequinum* | Chungnam (Nonsan) | Lung | + | |
| | b | | | | | Intestine | + | |
| Y4 | a | 2022-08-17 | Male | *R. ferrumequinum* | Chungnam (Nonsan) | Lung | + | OR572101 |
| | b | | | | | Intestine | + | OR761825 |
| Y5 | a | 2023-05-09 | Male | *R. ferrumequinum* | Gyungnam (Habcheon) | Lung | + | |
| | b | | | | | Intestine | + | |
| Y6 | a | 2023-05-20 | Male | *R. ferrumequinum* | Junbuk (Namwon) | Lung | + | OR761826 |
| | b | | | | | Intestine | + | OR761827 |
| Y7 | a | 2023-05-20 | Male | *R. ferrumequinum* | Junbuk (Sunchang) | Lung | + | |
| | b | | | | | Intestine | + | OR761828 |
| Y8 | a | 2023-05-20 | Male | *R. ferrumequinum* | Junbuk (Sunchang) | Lung | + | |
| | b | | | | | Intestine | + | |
| Y9 | a | 2023-05-09 | Male | *Miniopterus fuliginosus* | Gyungnam (Habcheon) | Lung | − | |
| | b | | | | | Intestine | − | |
| Y10 | a | 2023-05-20 | Male | *M. fuliginosus* | Junbuk; Namwon | Lung | − | |
| | b | | | | | Intestine | − | |
| Y11 | a | 2023-05-20 | Male | *M. fuliginosus* | Junbuk (Sunchang) | Lung | − | |
| | b | | | | | Intestine | − | |
| Y12 | a | 2023-05-20 | Male | *M. fuliginosus* | Junbuk (Sunchang) | Lung | − | |
| | b | | | | | Intestine | − | |
| Y13 | a | 2023-08-11 | Female | *R. ferrumequinum* | Gyungbuk (Yeongchun) | Lung | + | |
| | b | | | | | Intestine | + | |
| Y14 | a | 2023-08-11 | Female | *R. ferrumequinum* | Gyungbuk (Yeongchun) | Lung | + | |
| | b | | | | | Intestine | + | |
| Y15 | a | 2023-08-11 | Male | *Pipistrellus abramus* | Gyungbuk (Yeongchun) | Lung | − | |
| | b | | | | | Intestine | − | |
| Y16 | a | 2023-08-11 | Male | *P. abramus* | Gyungbuk (Yeongchun) | Lung | − | |
| | b | | | | | Intestine | − | |
| Y17 | a | 2023-08-11 | Female | *R. ferrumequinum* | Gyungbuk (Yeongchun) | Lung | + | |
| | b | | | | | Intestine | + | |
| Y18 | a | 2023-08-12 | Female | *Myotis petax* | Chungbuk (Danyang) | Lung | − | |
| | b | | | | | Intestine | − | |
| Y19 | a | 2023-08-12 | Female | *M. petax* | Chungbuk (Danyang) | Lung | − | |
| | b | | | | | Intestine | − | |
| Y20 | a | 2023-08-12 | Male | *M. aurascens* | Chungbuk (Danyang) | Lung | − | |
| | b | | | | | Intestine | − | |
| Y21 | a | 2023-08-12 | Male | *Myotis macrodactylus* | Chungbuk (Danyang) | Lung | − | |
| | b | | | | | Intestine | − | |
| Y22 | a | 2023-08-12 | Male | *M. macrodactylus* | Chungbuk (Danyang) | Lung | − | |
| | b | | | | | Intestine | − | |

sequences covering several genera (*alpha*, *epsilon*, *spuma-like*, and *lenti*; Fig. 3A), the results indicated that the complete genome data set (nt) presented the highest percentage of the sum of the three corner regions and the lowest percentage of the central region. The central region of the complete genome data set presented 10.4%, whereas those of other data sets (gag, pol, and env genes) exhibited overall high levels ranging from 23.1% to 28.3%, indicating poor reliability. This discrepancy is believed to

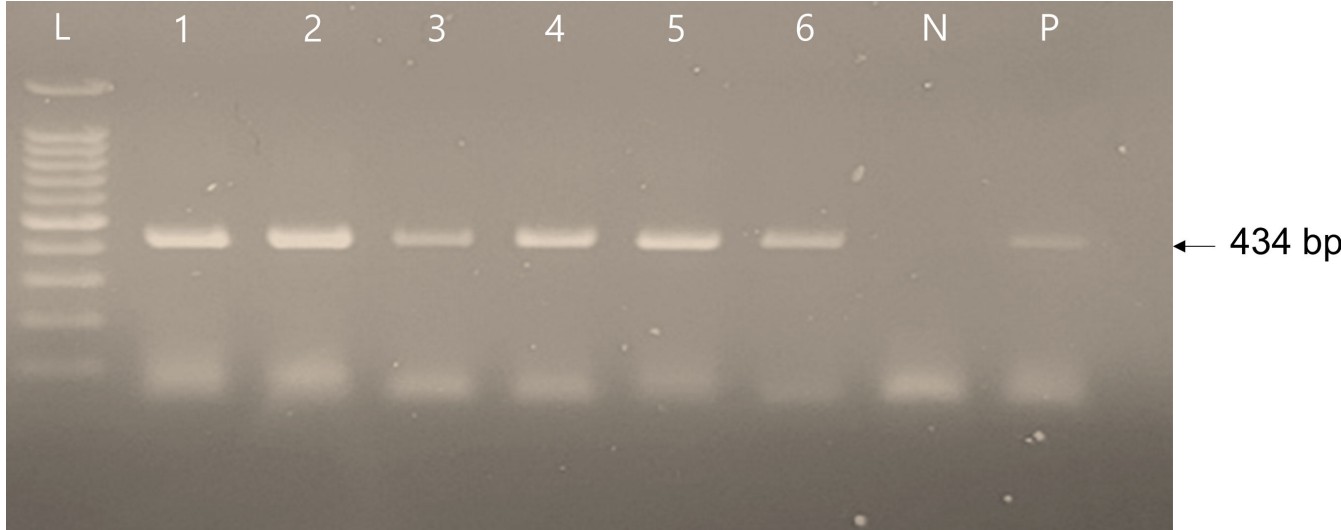

**FIG 1** Detection of RfRV in the tissue samples by RfRV-specific PCR. The lanes are labeled as follows: Lane L shows the 100 bp ladder, lanes 1–6 correspond to Y4a, Y4b, Y6a, Y6b, Y7a, and Y7b, respectively. Lane N represents the negative control, and Lane P represents the positive control.

be due to the significant genetic diversity among retroviral genera. Thus, the complete genome data set was considered the only reliable data set for topology analysis of retroviruses at the genus level. In the likelihood mapping analysis focused solely on *Gammaretroviruses* and bat retroviruses (Fig. 4), all data sets (the complete genome, gag, pol, and env) presented reasonably high levels in the sum of the three corner regions and low levels in the central region, although the complete genome data set presented the best resolution among the data sets. From the results, all data sets constructed in this study were confirmed to be reliable for use in topology analysis within *Gammaretrovirus*,

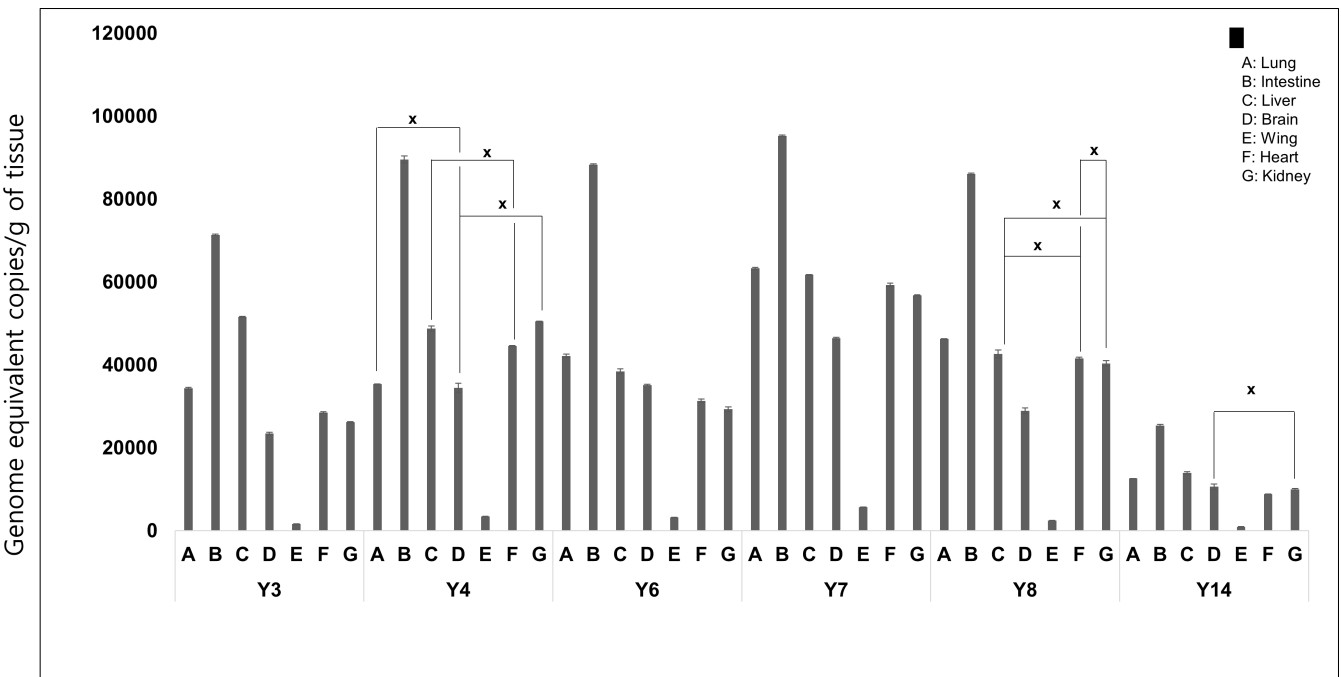

**FIG 2** Proviral loads of RfRV-positive samples. Quantification of proviral genome equivalent copies in the samples was performed by pol gene-specific dPCR. The viral load is presented in copies/g of tissues. The symbol "X" indicates no significance between the proviral loads of two samples in the paired *t* test ($P > 0.05$). Conversely, it indicates significance ($P < 0.05$) if otherwise.

A

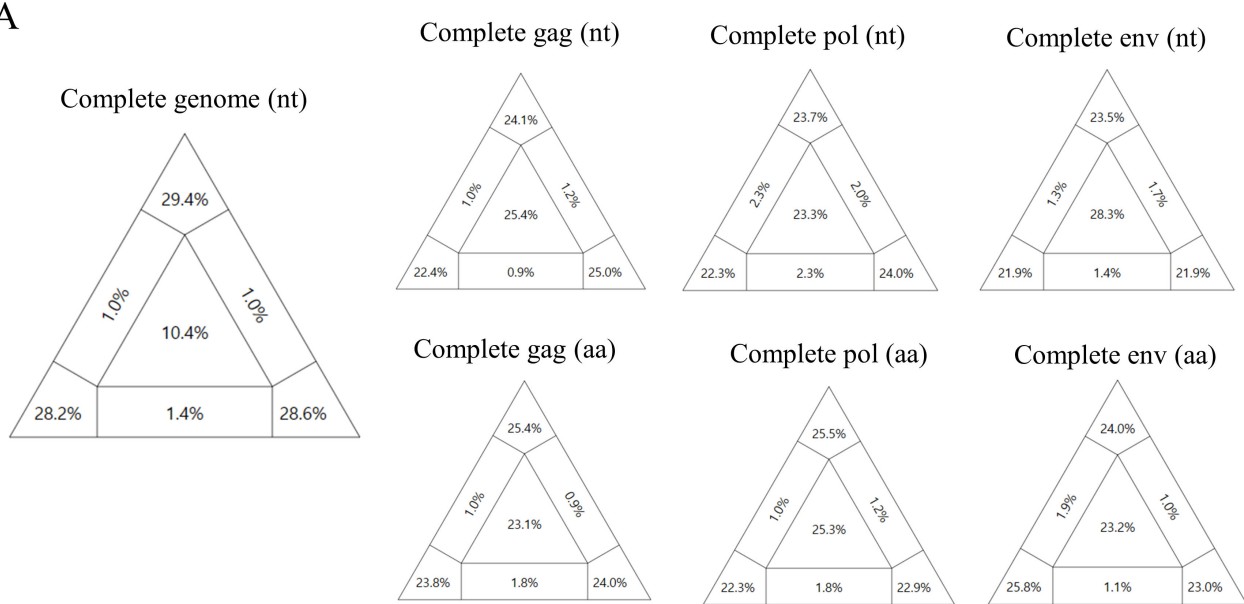

B

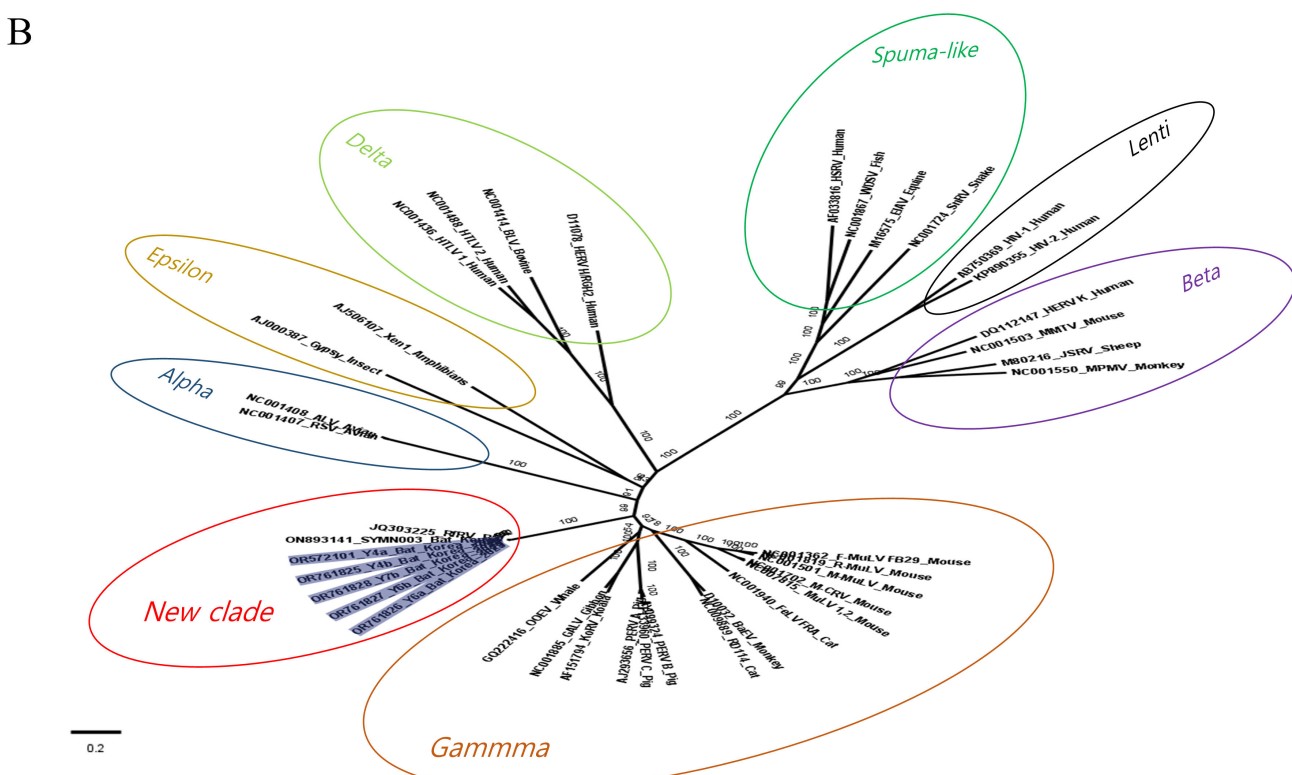

**FIG 3** The complete genome phylogeny of retroviruses. In panel A, likelihood mapping was applied to data sets comprising nucleotide sequences for three genes (gag, pol, and env) and complete genomes. The triangular plot presents corner values, indicating the percentage of well-resolved phylogenies for all possible quartets, while central and lateral values represent the percentages of unresolved phylogenies (noisy signal). Notably, the complete genome data set, selected as the best model, exhibits superior resolution compared to other mapping models. In panel B, Maximum Likelihood trees of complete genomes, with 1,000 bootstrap replicates, were generated using IQ-TREE, and the best-fitting model was automatically selected. South Korean strains are highlighted in gray, and strains Y4a, Y4b, Y6a, Y6b, and Y7b (from this study) are marked in red. Posterior-supported values are represented in the node bar.

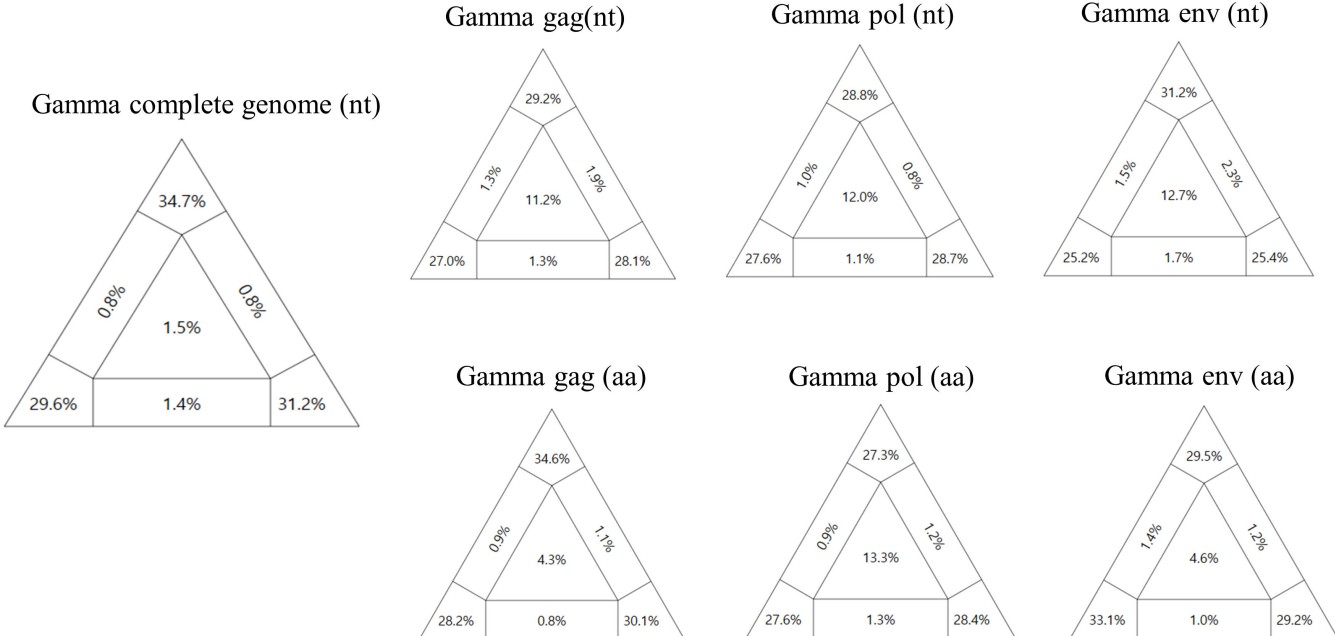

**FIG 4** Likelihood mapping on *Gammaretroviruses* and bat retroviruses. The analyses were applied to data sets for three genes (gag, pol, and env) and the complete genomes. The triangular plot presents corner values, indicating the percentage of well-resolved phylogenies for all possible quartets, while central and lateral values represent the percentages of unresolved phylogenies (noisy signal). All data sets presented reasonably high levels in the sum of the three corner regions and low levels in the central region, although the complete genome presented the best resolution.

and it was presumed that all data sets would be applicable for species-level topology analysis within a specific retroviral genus.

## Phylogenetic analysis of Korean bat retroviruses

The complete genome data set, confirmed as the best-fit model in the likelihood mapping analysis, was utilized for phylogenetic analysis on all collected retroviruses covering several genera. In the inferred phylogenetic tree, the Korean bat retroviruses formed a distinct new clade with RfRV (China, 2011) and SYMN003 (Kenya, 2022) strains (Fig. 3B).

In the phylogenetic analysis within *Gammaretrovirus*, all data sets, confirmed as reliable models from the likelihood mapping analyses, were utilized. The inferred phylogenetic trees from the complete genome (nt), gag (nt), and pol (nt and aa) data sets presented a distinct new clade including the Korean bat retroviruses, RfRV and SYMN003 (Fig. 5). However, in the inferred phylogenetic trees from the gag (aa) and env (nt and aa) data sets, the bat retroviruses belonged to the genus *Gammaretrovirus* (Fig. 6). Taken together, the phylogenetic topologies from multiple data sets were inconsistent. Nevertheless, we finally conclude that RfRV and RfRV-like viruses (the Korean bat retroviruses, RfRV and SYMN003) formed a new clade distinct from *Gammaretrovirus* according to the results from the complete genome data set (nt) because it was confirmed as the most reliable model through the likelihood mapping analyses. Additionally, RfRV-like viruses (or RfRV) presented a distinct, red-colored area in the heatmap derived from the pairwise comparison analysis, supporting the possibility of a new clade (Fig. S2). However, due to the limited number of RfRV or RfRV-like virus sequences available in this study, as little research on RfRVs has been conducted, only a few sequences were updated. To obtain more reliable results on the new clade, further study with a larger sample size is required.

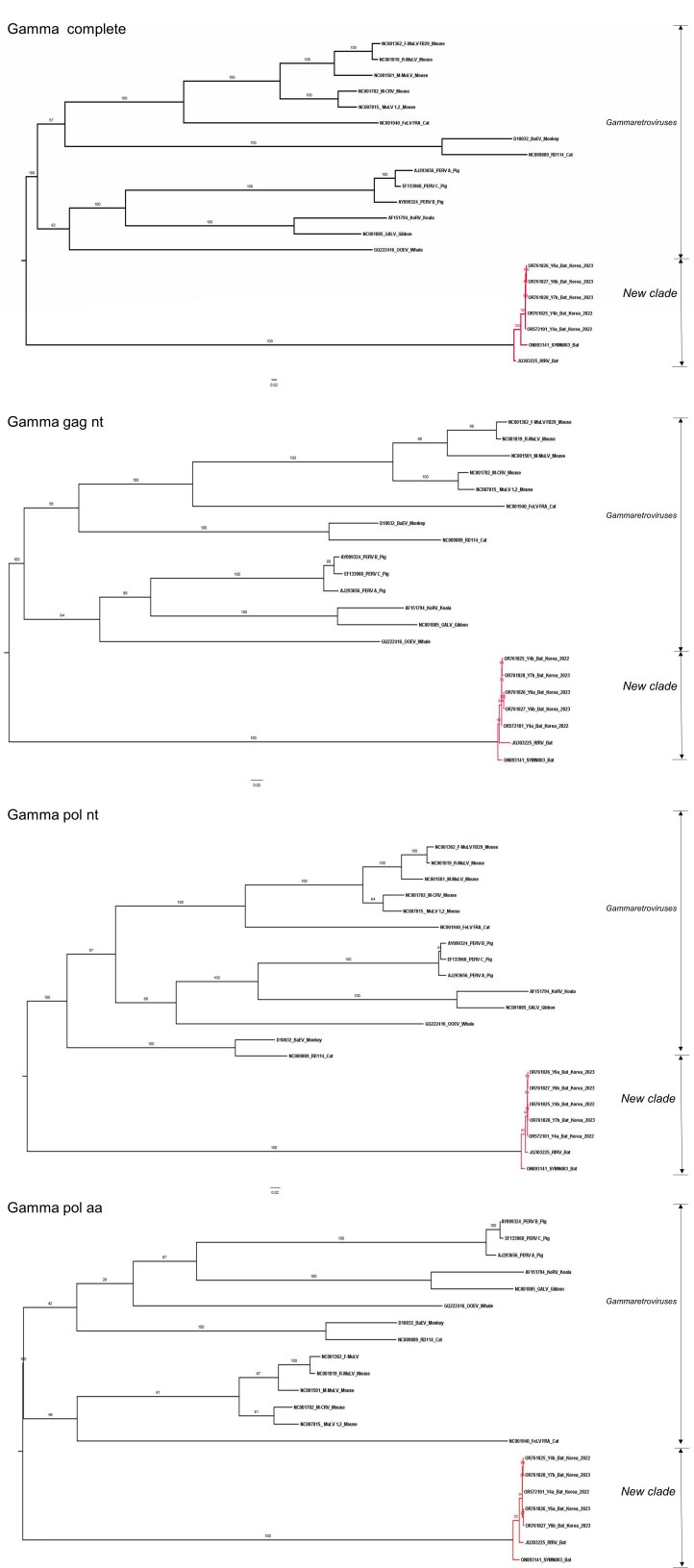

**FIG 5** The phylogenetic trees derived from the data sets of the complete genome (nt), gag (aa), and pol (nt and aa). Maximum Likelihood trees with 1,000 bootstrap replicates were generated using IQ-TREE, and the best-fitting model was automatically selected. In the inferred trees, the Korean bat retroviruses, RfRV and SYMN003, formed a new clade distinct from the genera *Gammaretrovirus*.

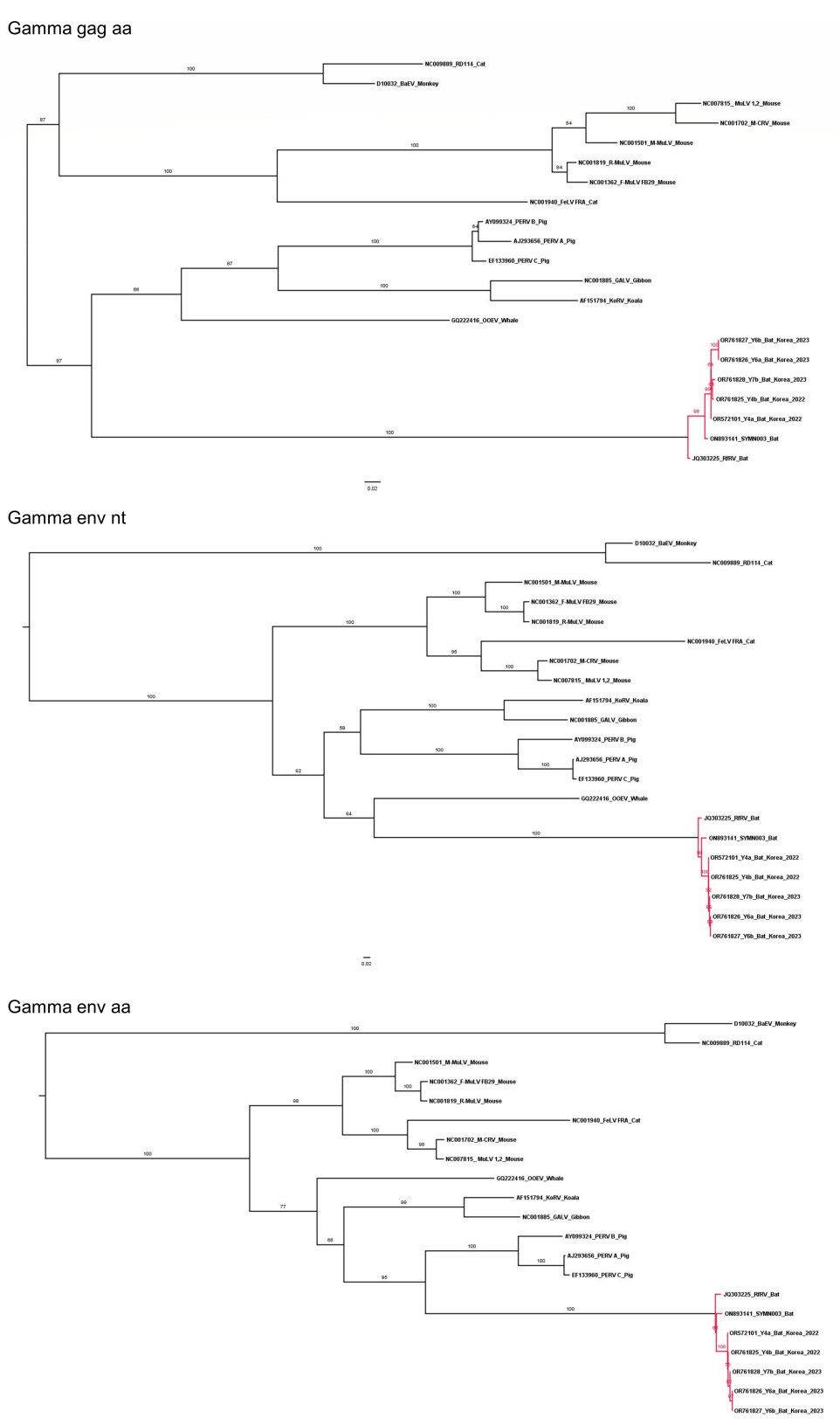

**FIG 6** The phylogenetic trees inferred from the data sets of gag (aa) and env (nt and aa). Maximum Likelihood trees with 1,000 bootstrap replicates were generated using IQ-TREE, and the best-fitting model was automatically selected. In the inferred trees, the Korean bat retroviruses, RfRV and SYMN003, belonged to the genera *Gammaretroviruses*.

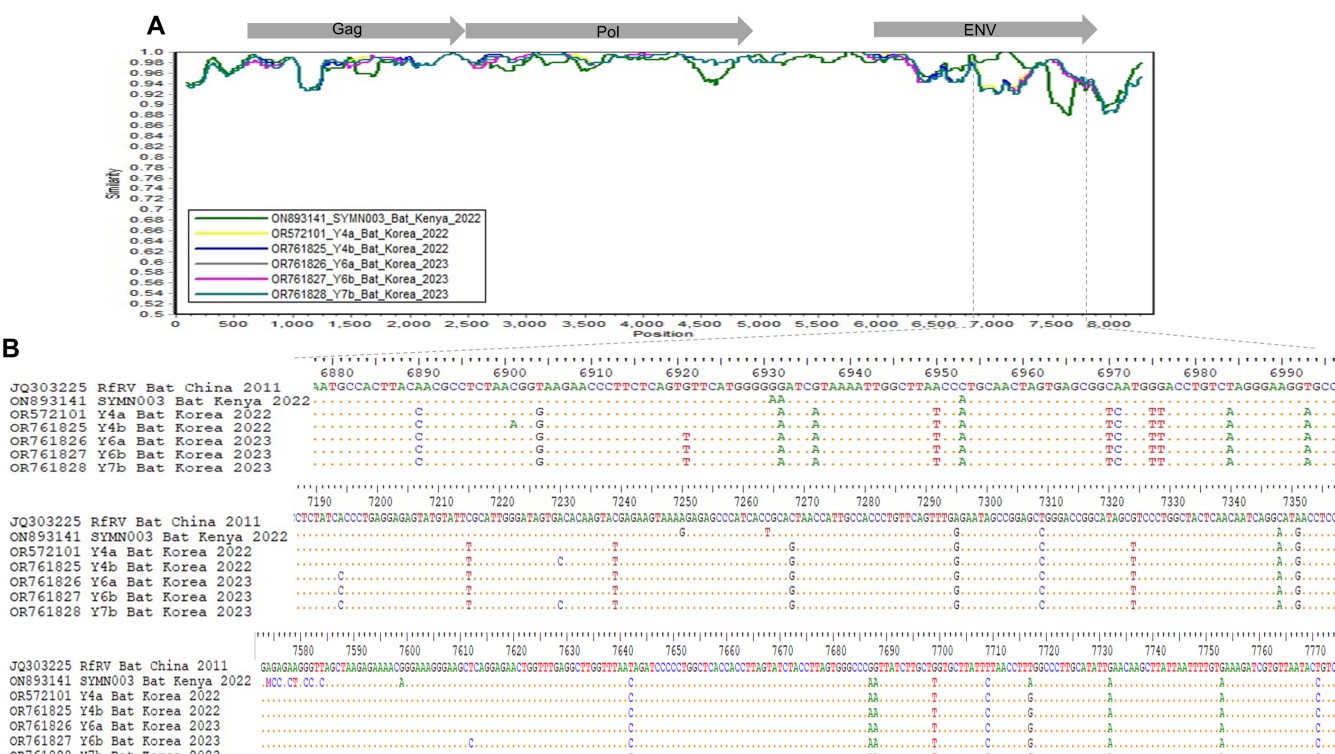

**FIG 7** Genetic comparison of RfRV strains. In Panel A, a Simplot comparison analysis between RfRV and RfRV-like viruses is presented, illustrating sequence similarities in the plot. Notably, a dashed line indicates the genomic region of the env gene where a drop in sequence similarity occurs. In Panel B, nucleotide comparisons are made specifically between the RfRV_Bat_China_2011 strain and other related strains. This analysis offers detailed insights into the genetic relationships among these strains.

## Genetic comparison between RfRV and RfRV-like viruses

In the Simplot comparison, RfRV-like viruses exhibited relatively strong fluctuating patterns in the latter part of the envelope (Env) gene area compared to other gene areas (Fig. 7A). Additionally, several point mutations within this region (6,878–7,774 bp) of these viruses were observed compared to the RfRV sequence (Fig. 7B). The viral envelope plays a crucial role in virus entry into the host cell by attaching to the viral receptor. Host immunity produces neutralizing antibodies against envelope proteins to protect against viral infection. These RfRV-like strains (2022–2023), discovered later than RfRV (2011), appear to be more evolved forms, and the accumulated mutations in the Env gene may allow reinfection of hosts already immunized to RfRV.

## Bat retrovirus isolation in cell lines

Virus recovery was attempted in several cell lines using the bat retrovirus-positive Y4b strain (from the Y4 intestine). Among the cell lines, the Y4b strain was successfully recovered in Raw 264.7 cells. During blind passage up to level 5, no cytopathic effects were exhibited in the cells. The viral titer steadily increased up to passage level 5 (Fig. 8A), and it was also confirmed that this virus robustly replicated over time, reaching 72 hours during incubation (Fig. 8B and C). Finally, virus particles released from the cells were observed by transmission electron microscopy (TEM), confirming virus isolation (Fig. 8D). The size of the mature virus particle was approximately 100 nm.

## DISCUSSION

Bats serve as reservoirs for numerous viruses across diverse viral families and are implicated in the transmission of many highly pathogenic viruses to various mammals,

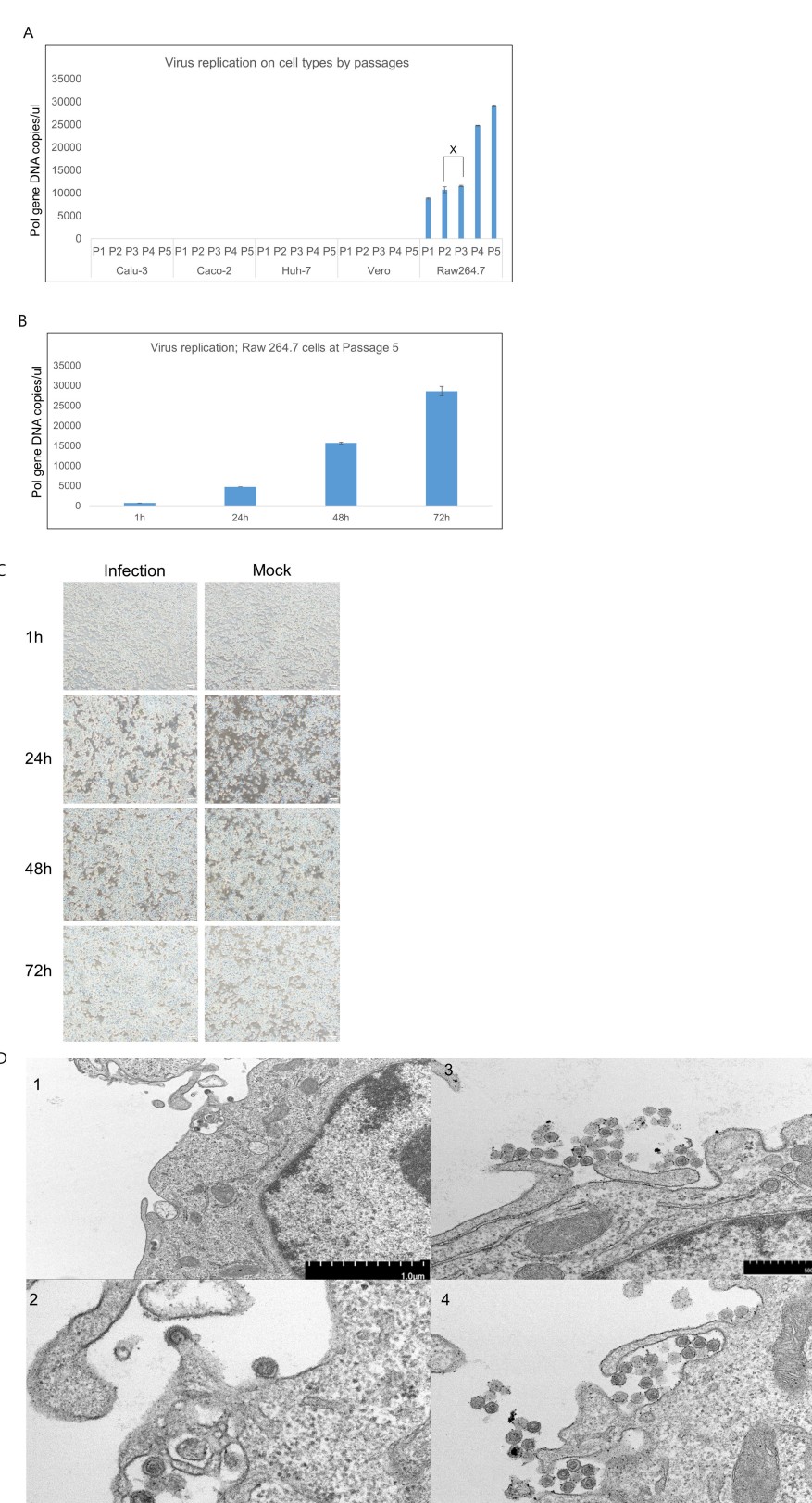

**FIG 8** Cell susceptibility examination of Y4b_Korea_2022 strain. In Panel A, human-originated cell lines (Calu-3, Huh-7, and Caco-2) and animal-originated cell lines (Vero and Raw 264.7) were infected with the virus at passages 1–5, and levels of cell supernatants were examined by qPCR. In Panel B, viral growth curves in passage 5 Raw 264.7 cell supernatants were examined (Continued on next page)

**FIG 8** (Continued)

by qPCR at 1, 24, 48, and 72 hours post-infection (hpi). Panel C showcases the results of the microscopic examination of Raw 264.7 cells infected by the Y4b_Korea_2022 strain at passage level 5, recorded at various time points (1, 24, 48, and 72 hours). In Panel D, TEM of Raw 264.7 cells was performed after inoculation with the Y4b_Korea_2022 strain at 48 hours. Number 1 depicts the presence of the virus within the Raw 264.7 cells and the shedding of the virus to the cells. Number 2 is a magnification of number 1, enlarged five times. Numbers 3 and 4 show an aggregation of viruses. The scale bar at the bottom of the black bar is indicated. "X" on the graph represents the non-significant group in the paired *t* test (*P* > 0.05), while experimental groups without symbols on the graph are considered significant (*P* < 0.05).

including humans (25). As the second-most species-rich order of mammals after rodents, bats (mammalian order *Chiroptera*) have 1,116 identified species worldwide (26). The unique features of bats, such as their ability to fly, aiding in virus dispersal and increasing horizontal transmission, their relatively long lifespan, their ability to hibernate over winter providing a mechanism for viruses to persist between seasons, and their genetic factors or unique immune systems that do not result in clinical disease, make them ideal reservoir hosts for viruses (27).

*R. ferrumequinum* (greater horseshoe bat) is an insectivorous bat belonging to the genus *Rhinolophus*, with a wide distribution covering Europe, Northern Africa, Central Asia, and Eastern Asia (28). RfRV was initially identified by Cui et al. (21) through transcriptome analysis using whole-brain samples from the greater horseshoe bat. RfRV-like viruses were also discovered in other *Rhinolophus* species, including *Rhinolophus pusillus*, *Rhinolophus pearsoni*, *Rhinolophus megaphyllus*, and *Rhinolophus affinis*, in China (22). In their study, these *Rhinolophus* retroviruses were classified as *Gammaretroviruses* based on the phylogenetic analysis using gag and pol amino acid sequences, and only the ERV form was confirmed, not the XRV form. Subsequently, Cui et al. (29) attempted to find evidence for harbored RfRV (or RFRV-like virus) in the genomes of its natural host (*R. ferrumequinum*) and nine other bat species (straw-coloured fruit bat, big brown bat, greater false vampire bat, Brandt's bat, David's myotis, little brown bat, black flying fox, Parnell's mustached bat, and large flying fox) through pan-phylogenomic analysis. However, they failed to detect the virus in these retroviruses (29). This result may be attributed to the limited sample size. In fact, the study utilized only one genome per bat species, retrieved from NCBI GenBank.

However, we succeeded in identifying five RfRV-like viruses (Y4a, Y4b, Y6a, Y6b, and Y7b) from greater horseshoe bats inhabiting in South Korea. Interestingly, these RFRV-like viruses formed a distinct new clade including RfRV (China, 2011) and SYMN003 (Kenya, 2022) strains in the phylogenetic analysis based on the full-length sequences. This is in contrast to the results of the phylogenetic analysis based on gag or pol proteins reported by Cui et al. (21), where it was concluded that RfRV belongs to the genus *Gammaretrovirus*. A typical retroviral genome consists of four canonical viral genes, gag (capsid protein), pro (viral protease), pol (replicative enzymes), and env (envelope protein) (5), and three major genes of gag, pol, and env have been commonly utilized for retroviral phylogenetic analyses (22, 30). Using as long sequences as possible is likely to elicit better results, as seen in the likelihood mapping analyses of this study. In the likelihood mapping analyses across several genera of retroviruses, the data sets of the three major genes (gag, pol, and env) exhibited significantly poor resolutions. Thus, only the complete genome was determined to be reliable for retroviral phylogenetic analysis at the genus level. In the subsequent phylogenetic analysis, a new clade of RfRV and RfRV-like viruses was identified. Whereas, in the likelihood mapping analyses focused on *Gammaretroviruses* and bat retroviruses, all data sets of the complete genome and three major genes were found to be reasonably reliable. Subsequently, all data sets were subjected to further phylogenetic analysis, but controversial results were obtained as the phylogenetic topologies from the data sets were inconsistent. In this study, we finally confirmed a new clade distinct from *Gammaretrovirus* according to the results from the complete genome data sets, confirmed as the most reliable data sets. Among the genes (gag, pol, and env), the phylogenetic topology from the pol data sets (nt and

aa) presented the most similar patterns to the complete genome data set (nt). Thus, this gene is believed to be the best option for retroviral phylogenetic analysis if analysis using the complete genome is not available due to difficulties in constructing or collecting whole-genome sequences.

Retroviruses are considered one of the most successful infectious pathogens, attributed to their extensive host range and the presence of millions of residual copies in various host genomes. Mammalian genomes contain abundant integrated sequences originating from ERVs, which are remnants of past integration events of XRVs (9, 31). XRVs facilitate horizontal transmission between inter- and intra-species, and the outcome of the disease depends on the interactions between the retrovirus and the host organism. When retroviruses infect germ line cells of the host, they can become permanent elements by transforming into ERVs in the host germ line, allowing for vertical transmission (32). Both ERVs and XRVs exist in *Betaretrovirus*, *Gammaretrovirus*, and *Deltaretrovirus* within the genomes of bats (21, 33, 34).

ERVs are typically not favored by natural selection, as most viral remnants present in host genomes are prone to inactivating mutations, truncation, or frequent insertion and deletion events due to genomic rearrangements. Consequently, ERVs that have been endogenized in the distant past are likely to lose their infectivity to other hosts. Many ERVs persist only as host-innocuous genetic remnants of their original XRVs (35). KoRV, a recently identified *Gammaretrovirus*, exists in both ERV and XRV forms. There are two major subtypes: KoRV-A and KoRV-B. KoRV-A is an infectious endogenous subtype capable of being an XRV, meaning it can be transmitted through both vertical and horizontal transmission. On the other hand, KoRV-B is an exogenous subtype that facilitates only horizontal transmission. The distinct feature of KoRV, with its subtypes KoRV-A and KoRV-B, suggests that the endogenization of KoRV in koalas is a relatively recent phenomenon (6, 32, 36). Similarly, the Korean RfRV-like viruses identified in this study were considered XRVs, similar to KoRV-B. The varying amounts of provirus detected in different organs suggest ongoing viral activity, replication, and *de novo* integration in certain organs. Furthermore, the successful recovery of Y4b in the Raw 264.7 cell line provides strong evidence supporting their status as XRVs. However, evidence for inheritance by vertical transmission was not found in this study, thus confirming the presence of ERVs from Korean RfRV-like viruses was not possible. Further study is required to confirm this aspect.

The potential for reinfection of host cells by infectious ERVs can be influenced by superinfection interference wherein host ERV envelope proteins block viral receptors, preventing subsequent reinfection by viruses with an envelope that targets the same receptor. Retroviruses have mechanisms to overcome superinfection interference, such as mutations in the viral envelope or recombination with other viral envelopes, leading to the emergence of XRVs with envelopes that utilize different receptors (32). In the similarity analysis between RfRV and RfRV-like viruses, the observed differences in homology in the Env gene suggest the potential for reinfection of host cells harboring RfRVs.

The presence of RfRVs (or RfRV-like viruses) has been revealed in various parts of the world over the past decade. Originally discovered in China in 2011, these viruses have now been identified in South Korea and, more recently, in Kenya. This geographic distribution spans Eastern Asia, Africa, and China, indicating a global presence of RfRV-like viruses (23). These viruses have been found in various bat species, including *R. pusillus*, *R. pearsoni*, *R. megaphyllus*, *R. affinis*, *R. landeri*, *Miniopterus natalensis*, and *Mycobacterium africanus*, as well as its natural host (*R. ferrumequinum*) (22, 23). It suggests that RfRV (or RfRV-like virus) facilitates cross-species transmission in the bat population. There is also evidence of transmission to other mammalian species. Cui et al. (29) revealed that non-bat ERVs extracted from pangolin and ferret genomes showed a high degree of relatedness with RfRV in the long terminal region alignment analysis (29).

Taken together, RfRV seems to have been introduced to the bat population relatively recently and is undergoing endogenization in real time. RfRV-like viruses are considered

XRVs originated from RfRV supported by the evidence revealed in this study, and these viruses currently appear to be spread worldwide since the first discovery of RfRV in 2011. Bats are the reservoir host of a wide diversity of potential zoonotic viruses, and these exogenous types of bat retroviruses may be one of them. Thus, the possibility of transmission to other animals beyond bats must be verified, and constant surveillance of these viruses is required.

In conclusion, five RfRV-like viruses (Y4a, Y4b, Y6a, Y6b, and Y7b strains) were identified in greater horseshoe bats inhabiting in South Korea. These RfRV-like viruses were presumed to be XRVs, supported by the following results: successful virus recovery in cell lines and varying proviral loads in various organs. Based on previous and recent data, it appears that RfRV was introduced to the bat population relatively recently and is currently undergoing endogenization. Additionally, RfRV-like viruses have spread worldwide since the first discovery of RfRV in 2011.

## MATERIALS AND METHODS

### Sample collection

From September 2022 to August 2023, bat carcasses from a total of six species [*Myotis (M) aurascens*, *M. petax*, *M. macrodactylus*, *Miniopterus fuliginosus*, *R. ferrumequinum*, and *Pipistrellus abramus*] were collected from various areas across South Korea (refer to Table 1). The bat species were confirmed by a bat ecology expert (Prof. Chul-Un Chung participated in this study) through external features such as size, color, and ear shape, as well as anatomical features including skull structure, tooth shape, and wing structure (37). The collected carcasses, stored at 4°C, were subjected to the College of Medicine, Yonsei University at Biosafety level 3 (BL3) in the Avison Biomedical Research Center and approved by the Institutional Biosafety Committee (IBC 2022–0320). After the necropsy of the bats, various organs (lung, intestine, heart, brain, wing, kidney, and liver) were collected, and 10 mg of each tissue sample was homogenized in 10 mL of Dulbecco's Modified Eagle Medium (DMEM). The homogenized samples were preserved at −70°C for further study.

### DNA extraction

For DNA extraction, a mixture of 8 µL of Proteinase K solution, 500 µL of DNA lysis buffer, and 200 µL of samples were used. The mixture was thoroughly vortexed and incubated for 1 hour. Subsequently, 200 µL of phenol–chloroform–isoamyl alcohol (25:24:1) was added, thoroughly vortexed, and centrifuged at 13,000 rpm for 10 min. The DNA in the aqueous phase was precipitated with an equal volume of isopropanol, followed by centrifugation. The resulting DNA pellet was washed with 1 mL of 70% ethanol, centrifuged, dried, and re-suspended in 30 µL Tris-EDTA (TE) buffer. The DNA concentrations were measured using the NanoDrop spectrophotometer 2000 (Thermo Scientific). The final concentration of DNA was adjusted to 10 ng/µL for PCR and dPCR assays.

### Detection and quantification of retrovirus in bats

The detection and quantification of bat retrovirus were performed by PCR and dPCR assays employing specific primers designed in this study. Due to the limited availability of bat retrovirus sequences in GenBank, only two bat retrovirus sequences (access no. JQ303225 and ON893141) were utilized as input data for primer selection. The sequences used for primer design and primer information are provided in Table S1.

First, conventional PCR was conducted to detect bat retrovirus. For PCR, the reaction mixture comprised 2 µL of template DNA, 1 µL of each primer (10 µM), and 16 µL of Master mix solution (Intron Biotech, Korea). The thermal profile included an initial denaturation for 5 min at 95°C, followed by 40 cycles of 95°C for 30 s, 58°C for 30 s, 72°C for 30 s, and a final extension for 7 min at 72°C. PCR products were analyzed through

electrophoresis on a 1% agarose gel with Red Safe DNA nucleic acid staining solution (iNtRON Biotechnology, Inc. Korea).

Subsequently, the quantification of provirus in various organs was performed by dPCR assay. For dPCR, a reaction mixture of 30 µL included 5 µL of primer–probe mixture (Supplementary Table; 20 pmol forward primer, 20 pmol reverse primer, and 10 pmol FAM-BHQ1 probe per reaction), 15 µL of 2× Dr. PCR premix, 10 ng DNA, and nuclease-free water up to 30 µL. The reaction mixtures were loaded into wells of LOAA Dr. Digital PCR cartridges (Optolane, Korea). The cartridges were placed into the POSTMAN equipment (Optolane, Korea) for uniform application and then mounted on the LOAA equipment (Optolane, Seongnam-si, Korea). The digital PCR employed a two-step cycling protocol: a Uracil-DNA Glycosylase step of 3 min at 50°C, an initial denaturation step of 15 min at 95°C, followed by 40 cycles of 95°C for 10 s and 60°C for 15 s. Each sample produced 16,800–19,200 valid wells. Digital PCR results were analyzed using the "Optolane OnPoint Pro" software (Optolane, Korea). A primer working test for bat retrovirus detection was conducted using equivalent copies, guided by a standard curve constructed from plasmids ($1.12 \times 10^5$ copies/µL) containing the cloned retrovirus sequence of 350 bp, with dilutions ranging from $10^0$ to $10^{-4}$. The viral loads were log10-transformed and normalized to the dilution factor (Fig. S1).

## Complete genome amplification and sequencing

Among the positive samples, full-length sequences of five strains (Y4a, Y4b, Y6a, Y6b, and Y7b) were constructed through the primer-walking method with seven pairs of specific primers (see Table S). The specific PCR products were purified through gel extraction and subsequently processed for T4 DNA polymerase sequence-and ligation-independent (TA) cloning and transformation. Sanger sequencing was conducted by Macrogen Inc. (Seoul, South Korea). The complete genomes of the five bat retrovirus strains have been deposited into the NCBI GenBank with accession numbers OR572101 and OR761825 to OR761828.

## Likelihood mapping

To determine the optimal phylogenetic tree model, three protein-coding genes (gag; 1728nt, pol; 2586nt, env; 1878nt) and the complete genome (8363nt) of retroviruses were chosen for likelihood mapping. Reference sequences retrieved from NCBI GenBank (Table S2), including information on the host's origin, were incorporated into the analysis. Recombinants of the collected retrovirus sequences were removed using recombination detection program (RDP) version 4.101 (38) prior to further analysis. The final data set comprised 39 retrovirus sequences originating from animals, humans, fish, and insects, with each of the three coding genes aligned using the MAFFT L-INS-i method (39). The phylogenetic signal of the sequences was assessed through likelihood mapping analysis using IQ-TREE version 1.3.8 (40). The selection of the best nucleotide substitution likelihood mapping model was based on option mapping bootstrap 1,000 (-lmap 1,000). For each alignment, the model was automatically determined by specifying the "-m TESTONLY" option. The analysis visualized the phylogenetic content of aligned sequences by plotting probability vectors of any quartets of taxa in an equilateral triangle. Results were presented with a partitioned triangular graph, where the three tips of the triangle represented the resolved percentages of quartets, and three rectangles on the sides represented quartets with network evolution (conflicting signal). The central region of the triangle represented star-like evolution (noisy signal).

## Phylogenetic analysis

The best model was chosen based on the central value (low noise) and the sum of bootstrap supporting values from three triangular peaks (41, 42). The best nucleotide substitution model was automatically selected using the "-m TEST" option in IQ-TREE version 1.3.8 (40), and an inferred phylogenetic tree was extracted.

## Genetic comparison of bat retroviruses

Comprehensive genome analysis was conducted on the whole-genome sequences of bat retroviruses retrieved from NCBI GenBank or identified in this study. Pairwise sequence identities were calculated based on multiple sequence alignments using SDTv1.2 (43). The sliding window analysis, implemented in SimPlot (44), was employed to visualize the similarity vs position across the whole-genome sequences of bat retroviruses, with RfRV chosen as the query sequence in the analysis.

## Bat retrovirus isolation

Human hepatoma cells (Huh-7), African green monkey kidney cells (Vero), murine macrophage cells (Raw 264.7), human bronchial submucosal gland cells (Calu-3), and human colorectal adenocarcinoma tissue cells (Caco-2) were cultured in DMEM supplemented with 10% fetal bovine serum (FBS) (growth media). Maintenance media for these cells consisted of the same DMEM supplemented with 0.3% tryptose phosphate broth, 0.02% yeast extract, and 1 μg/mL trypsin. All cells were maintained at 37°C and 5% $CO_2$.

To investigate the cell tropism of the bat retroviruses identified in this study, tissue samples (titer; 5,000 copies/μL) were inoculated into the cell lines for 1 hour at 37°C to allow absorption. Subsequently, cells in 48 wells ($5 \times 10^5$ cells) were washed twice with phosphate-buffered saline (PBS) and cultured in either growth medium or maintenance medium for 72 hours, spanning serial passages from levels 1 to 5. To investigate the level of virus growth, the viral load at each cell passage was quantified in triplicate using dPCR. Upon confirming virus growth in the cells, virus quantification was performed at different time points (1, 24, 48, and 72 hours) after virus inoculation at passage level 5. Viral load at each time-point was quantified in triplicate using dPCR. All virus quantification was performed with cell culture supernatants.

## Retrovirus electron microscopy

Viral supernatant at passage level 5 was inoculated onto confluent cells in a25T flask and incubated for 72 hours. For the observation of the virus using TEM, viral-inoculated Raw 264.7 cells from the 25T flask were harvested at 2500 rpm at 4°C for 10 min. The virus was further concentrated using an Amicon 50 kDa filter (Merck Millipore, Germany), followed by fixation with 0.1% formaldehyde for 4 hours. Subsequently, cells were washed, dehydrated using a graded ethanol series (50%, 60%, 70%, 80%, 90%, and 100%), and infiltrated with an Embed-812 embedding kit (Electron Microscopy Sciences, US). The embedding was polymerized in an oven at 60°C for 48 hours. Samples were negatively stained and examined using HT7800 TEM (Hitachi High-tech, Tokyo, Japan).

## Statistical analysis

All experiments were repeated three times. Statistical comparisons were performed (Fig. 2, 8A and B) using paired $t$ tests ($P < 0.05$) within the SPSS program (version 23.0.0; SPSS Inc., USA). In the graphs, an "X" indicates the non-significant group in the paired $t$ test ($P > 0.05$), while experimental groups without symbols on the graph are considered significant ($P < 0.05$).

## ACKNOWLEDGMENTS

The authors would like to thank Sung Hoon Park and Hyeon Woo Chung for excellent technical assistance. This work was supported by the Korea Science and Engineering Foundation (KOSEF) grant (No. 2020R1I1A1A01054539) and by the Korea Research Foundation Grant funded by the Korean Government (NFR-2019R1A6A1A03032869).

H.C.C.: methodology, formal analysis, supervision, writing—original draft, and writing—review and editing. S.J.K.: data curation, methodology, validation, visualization, writing—original draft, and formal analysis. S.J.H.: resources and methodology. Y.S.J.:

resources. M.S.S.: methodology, formal analysis, and resources. S.H.K.: Methodology. J.L.: writing—review and editing. Y.C.: methodology. C.U.C.: conceptualization, resources, and validation. J.M.L.: conceptualization, writing—review and editing, funding acquisition, and project administration.

The funders had no role in the design of the study; in the collection, analyses, or interpretation of data; in the writing of the manuscript; or in the decision to publish the results.

## AUTHOR AFFILIATIONS

[1]Department of Microbiology and Immunology, Institute for Immunology and Immunological Diseases, Yonsei University College of Medicine, Seoul, South Korea
[2]Department of Companion Animal Health, Kyungbok University, Namyangju, South Korea
[3]Department of Microbiology and Immunology, Institute for Immunology and Immunological Diseases, Brain Korea 21 Project for Medical Science, Yonsei University College of Medicine, Seoul, South Korea
[4]Department of Life Science, Dongguk University, Gyeongju, South Korea
[5]Bio Institute, OPTOLANE Technologies Inc, Seongnam-si, Gyeonggi-do, South Korea
[6]Department of Microbiology, California University of Science and Medicine, Colton, California, USA

## AUTHOR ORCIDs

Hee Chun Chung  http://orcid.org/0000-0003-4666-5393
Sung Jae Kim  http://orcid.org/0000-0002-8813-012X
Su Jin Hwang  http://orcid.org/0000-0001-9182-8281
Young Shin Jeon  http://orcid.org/0000-0001-9015-3681
Min Sik Song  http://orcid.org/0000-0002-0658-5666
Si Hwan Ko  http://orcid.org/0009-0005-3739-4597
Jasper Lee  http://orcid.org/0000-0003-2086-1441
Yoona Choi  http://orcid.org/0009-0001-8191-4909
Chul Un Chung  http://orcid.org/0000-0002-7283-8668
Jae Myun Lee  http://orcid.org/0000-0002-5273-3113

## FUNDING

| Funder | Grant(s) | Author(s) |
| --- | --- | --- |
| Korea Science and Engineering Foundation (KOSEF) | 2020R1I1A1A01054539 | Hee Chun Chung |
| Korea Resources Corporation (KORES) | NFR-2019R1A6A1A03032869 | Jae Myun Lee |

## AUTHOR CONTRIBUTIONS

Hee Chun Chung, Formal analysis, Methodology, Supervision, Writing – original draft, Writing – review and editing | Sung Jae Kim, Data curation, Formal analysis, Methodology, Validation, Visualization, Writing – original draft | Su Jin Hwang, Methodology, Resources | Young Shin Jeon, Resources | Min Sik Song, Formal analysis, Methodology, Resources | Si Hwan Ko, Methodology | Jasper Lee, Writing – review and editing | Yoona Choi, Methodology | Chul Un Chung, Conceptualization, Resources, Validation | Jae Myun Lee, Conceptualization, Funding acquisition, Project administration, Writing – review and editing

## DATA AVAILABILITY

All data generated or analyzed during this study are included in this published article and its additional files. The data sets analyzed in the current study are available from the

corresponding author upon reasonable request. The complete genomes of the five bat retrovirus strains have been deposited into the NCBI GenBank with accession numbers OR572101 and OR761825 to OR761828.

## ETHICS APPROVAL

This article does not contain any studies with alive animals performed by any of the authors.

## ADDITIONAL FILES

The following material is available online.

### Supplemental Material

**Supplemental Figure 1 (Spectrum04323-23-s0001.pdf).** Regression analysis.
**Supplemental Figure 2 (Spectrum04323-23-s0002.pdf).** Presents heatmaps.
**Supplemental material (Spectrum04323-23-s0003.pdf).** This study sequences.
**Supplemental Table 1 (Spectrum04323-23-s0004.pdf).** List of primers used in this study.
**Supplemental Table 2 (Spectrum04323-23-s0005.pdf).** This sequences used in tree analysis.

### Open Peer Review

**PEER REVIEW HISTORY (review-history.pdf).** An accounting of the reviewer comments and feedback.

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
