## [Reviewer comments · Microbiology Spectrum]

Microbiology Spectrum

Identification and Characterization of Recent Endogenous Retrovirus in *Rhinolophus ferrumequinum* Bats

HeeChun Chung, SungJae Kim, SuJin Hwang, Young Shin Jeon, MinSik Song, SiHwan Ko, Jasper Lee, Yoona Choi, Chul-un Chung, and Jae Myun Lee

Corresponding Author(s): HeeChun Chung, Yonsei University College of Medicine

Review Timeline:

Submission Date:	December 31, 2023
Editorial Decision:	February 8, 2024
Revision Received:	March 21, 2024
Accepted:	March 25, 2024

Editor: Takamasa Ueno

Reviewer(s): The reviewers have opted to remain anonymous.

Transaction Report:

DOI: <https://doi.org/10.1128/spectrum.04323-23>

Re: Spectrum04323-23 (Identification and Characterization of Recent Endogenous Retrovirus in Rhinolophus ferrumequinum Bats)

Dear Dr. HeeChun Chung:

Thank you for the privilege of reviewing your work. Below you will find my comments, instructions from the Spectrum editorial office, and the reviewer comments.

I received comments from the two experts and both found some interests in the current version although additional clarification is needed. In particular, the phylogenetic analyses should be improved as pointed out by the reviewers.

Revision Guidelines

Sincerely,
Takamasa Ueno
Editor
Microbiology Spectrum

Reviewer #1 (Comments for the Author):

Chung et al. analyzed retroviruses in several bat samples and identified RfRV-like viruses in Rhinolophus ferrumequinum samples. They obtained complete genome sequences of RfRV-like viruses, and by conducting phylogenetic analyses, RfRV-like and RfRV viruses are distinct from other gammaretroviruses. Further, they isolated infectious RfRV-like viruses in cell lines.

The study contains various interesting results; however, there are many gaps in the results described in the text, especially in the phylogenetic analyses. The comments below could be helpful to improve the manuscript.

I cannot find any clear evidence that RfRV-like viruses are endogenous retroviruses. Did the authors examine the genomic location of five RfRV-like viruses (Y4a, Y4b, Y6a, Y6b, and Y7b) identified in this study and some completely identical?

The logic of detection and quantification of bat retroviruses noted in this manuscript needs to be clarified. The authors conducted PCR and found bat retroviruses only in *Rhinolophus ferrumequinum*'s samples. Then, dPCR using genomic DNA of seven tissue samples identified that the copy numbers varied depending on the tissues, even in the same individuals. Why did the authors conduct RNA-seq and/or qPCR using RNA samples of each tissue? If this is the case, more direct evidence can be shown.

The results of the likelihood mapping analysis could be clearer for me. They finally claimed in the section, "The complete genome dataset was determined to be the best fit model in topology analysis among the 4 datasets tested."; however, it is generally known that the longer sequences could be more informative without any tests. The authors should care about the possible recombination(s) in the retroviral sequences used for the phylogenetic analyses. The authors should examine the possibility, and if this is the case, the phylogenetic analysis should care about the recombination breaks.

According to their phylogenetic results, "The p-distances of the bat retrovirus intra-group [0.0004-0.0268] and bat retroviruses vs gammaretroviruses inter-group [0.7019 - 0.8721] formed clearly discrete areas" suggests that the RfRV and RfRV-like viruses are not well aligned with other gammaretroviruses used in this study. Considering the diversity of the gammaretroviruses, the result itself is not surprising. The authors should use amino acid sequences of RT and TM regions of Pol and Env, respectively, for the phylogenetic analyses because they are known to be conserved relatively.

Line #23: "(R)" is not needed.

Line #78: "Jie et al." should be "Cui et al."

Line #255: "(M)" is not needed.

Reviewer #2 (Comments for the Author):

1. In line 39 of the abstract, the author wrote the following sentence: "and the Korean RfRV-like viruses were assumed to be ERVs that evolved from RfRV." I suggest revising this statement in the abstract. The author mentions in the discussion section that there are few results to confirm this theory, therefore, this statement in the abstract may lead to misunderstandings about the work.
2. How do the authors classify the taxonomy of the bats? Please add this to the methodology section. Who identifies the species and which methods are used for species identification (either molecular or otherwise)?
3. Please describe in detail all measures to prevent contamination, e.g., whether there is a dedicated PCR room, which controls were used in the extraction and dPCR procedures, etc.
4. Please describe in detail all criteria used for selecting the public sequence database used for phylogenetic inference, e.g., all complete sequences deposited up to period X in the NCBI.
5. About the cell tropism experiment, it's unclear how many replicates were utilized. From how I've interpreted this, a single culture was conducted for each cell and tissue type (considered a single biological replicate), while triplicates were carried out for the dPCR assay (considered a technical replicate). I believe that even with statistical evaluation for the dPCR triplicates, it may not be possible to assess the difference between tissues, considering that a greater deviation would likely be observed in culture replicates.
6. Could you provide more details about the discussion on the cellular tropism experiment?

Identification and Characterization of Recent Endogenous Retrovirus in *Rhinolophus ferrumequinum* Bats

Comments and Suggestions:

1. In line 39 of the abstract, the author wrote the following sentence: “and the Korean RfRV-like viruses were assumed to be ERVs that evolved from RfRV.” I suggest revising this statement in the abstract. The author mentions in the discussion section that there are few results to confirm this theory, therefore, this statement in the abstract may lead to misunderstandings about the work.
2. How do the authors classify the taxonomy of the bats? Please add this to the methodology section. Who identifies the species and which methods are used for species identification (either molecular or otherwise)?
3. Please describe in detail all measures to prevent contamination, e.g., whether there is a dedicated PCR room, which controls were used in the extraction and dPCR procedures, etc.
4. Please describe in detail all criteria used for selecting the public sequence database used for phylogenetic inference, e.g., all complete sequences deposited up to period X in the NCBI.
5. About the cell tropism experiment, it's unclear how many replicates were utilized. From how I've interpreted this, a single culture was conducted for each cell and tissue type (considered a single biological replicate), while triplicates were carried out for the dPCR assay (considered a technical replicate). I believe that even with statistical evaluation for the dPCR triplicates, it may not be possible to assess the difference between tissues, considering that a greater deviation would likely be observed in culture replicates.
6. Could you provide more details about the discussion on the cellular tropism experiment?

We sincerely appreciate the efforts of editor and reviewers. We revised the manuscript after discussion according to the reviewer's comments and we have revised the manuscript, resulting in a significantly improved version. Below, you will find our opinions and revisions in response to the reviewer's comments.

Reviewer #1 (Comments for the Author):

1. I cannot find any clear evidence that RfRV-like viruses are endogenous retroviruses. Did the authors examine the genomic location of five RfRV-like viruses (Y4a, Y4b, Y6a, Y6b, and Y7b) identified in this study and some completely identical?

Answer) I agree with the reviewer's opinion. There was an error in the interpretation of our results. We proposed that these viruses are infectious ERVs convertible to XRVs. However, we could not find any evidence for endogenization, such as inheritance by vertical transmission, which is a major feature of ERVs. Thus, in this study, we concluded that the Korean RfRV-like viruses are XRVs and revised the related contents **in the abstract, importance, results (lines 89-101), and discussion (lines 236-243, 270-276).**

We succeeded in recovering the full-length sequences of five strains (Y4a, Y4b, Y6a, Y6b, and Y7b) and registered these sequences (currently in private status) in NCBI GenBank. We have sent the information as an attachment (file titled **'This study sequences information.pdf'**)

2. The logic of detection and quantification of bat retroviruses noted in this manuscript needs to be clarified. The authors conducted PCR and found bat retroviruses only in *Rhinolophus ferrumequinum*'s samples. Then, dPCR using genomic DNA of seven tissue samples identified that the copy numbers varied depending on the tissues, even in the same individuals. Why did the authors conduct RNA-seq and/or qPCR using RNA samples of each tissue? If this is the case, more direct evidence can be shown.

Answer) I apologize for the lack of explanation regarding the use of two types of PCR in this study. Initially, conventional PCR was employed to detect retroviruses in the bats. However, we encountered difficulty in determining whether the detected viruses were ERVs or XRVs, as they were detected solely from DNAs extracted from the tissue samples via PCR. Consequently, we conducted digital PCR (dPCR) on seven types of tissue samples derived from each individual to obtain evidence for determining the type of these viruses.

Our hypothesis was that the proviral loads in tissue samples would be similar if the detected virus was not in an active form (ERV). This concept was inspired by Hashem et al.'s study (2020). The explanation for this methodology was detailed in the 'Detection of Bat Retrovirus' section of the results (lines 304-330) (lines 304-330).

* Hashem *et al.*, 2020. Transmission of koala retrovirus from parent koalas to a joey in a Japanese zoo. *Journal of virology* 94(11), 10.1128/jvi.00019-00020.

3. The results of the likelihood mapping analysis could be clearer for me. They finally claimed in the section, "The complete genome dataset was determined to be the best fit model in topology analysis among the 4 datasets tested."; however, it is generally known that the longer sequences could be more informative without any tests. The authors should care about the possible recombination(s) in the retroviral sequences used for the phylogenetic analyses. The authors should examine the possibility, and if this is the case, the phylogenetic analysis should care about the recombination breaks.

Answer) I agree with your comment regarding the possibility of recombination events potentially interfering with phylogenetic analysis. To address this concern, we identified and removed recombinants within all sequences used in this study using the Recombination Detection Program (RDP) prior to further genetic or phylogenetic analyses (lines 346-347).

4. According to their phylogenetic results, "The p-distances of the bat retrovirus intra-group [0.0004-0.0268] and bat retroviruses vs gammaretroviruses inter-group [0.7019 - 0.8721] formed clearly discrete areas" suggests that the RfRV and RfRV-like viruses are not well aligned with other gammaretroviruses used in this study. Considering the diversity of the gammaretroviruses, the result itself is not surprising. The authors should use amino acid sequences of RT and TM regions of Pol and Env, respectively, for the phylogenetic analyses because they are known to be conserved relatively.

Answer) As per your comment, we have removed the p-distance result from the manuscript as it was deemed not surprising or worth discussing.

Regarding the suggestion regarding phylogenetic analysis, it's worth noting that many previous studies have utilized various datasets, including the complete genome, complete sequences, or partial parts of major genes, depending on research conditions such as the possibility of recovering or collecting sequences. For instance, Hoyward et al. (2020) and Cui et al. (2012) employed datasets based on nucleotide (nt) or amino acid (aa) sequences of the

complete genome or complete sequences of three major genes (gag, pol, and env) for retroviral phylogenetic analyses in their studies.

At present, I am uncertain whether the phylogenetic analysis necessarily needs to be conducted using the suggested regions, specifically the reverse transcriptase (RT) or transmembrane (TM) regions of the pol and env genes. Additionally, these regions may be relatively short to yield optimal results. Therefore, we opted for a more comprehensive approach, conducting detailed analyses on the complete genome as well as the complete sequences of the gag, pol, and env genes for likelihood mapping and phylogenetic analysis. The results and corresponding discussions have been updated and included in the revised manuscript (lines 104- 123, 126-145, 197-216).

* Hoyward *et al.*, Infectious KoRV-related retroviruses circulating in Australian bats. 2020. Proceedings of the National Academy of Sciences 117(17), 9529–9536.

* Cui *et al.*, 2012b. Discovery of retroviral homologs in bats: implications for the origin of mammalian gammaretroviruses. Journal of Virology 86(8), 4288-4293.

Line #23: "(R)" is not needed.

Answer) The words pointed out were deleted or revised.

Line #78: "Jie et al." should be "Cui et al."

Answer) The words pointed out were deleted or revised.

Line #255: "(M)" is not needed.

Answer) The words pointed out were deleted or revised.

Reviewer #2 (Comments for the Author):

1. In line 39 of the abstract, the author wrote the following sentence: "and the Korean RfRV-like viruses were assumed to be ERVs that evolved from RfRV." I suggest revising this statement in the abstract. The author mentions in the discussion section that there are few results to confirm this theory, therefore, this statement in the abstract may lead to misunderstandings about the work.

Answer) I agree with the reviewer's opinion. There was an error in the interpretation of our results. We proposed that these viruses are infectious ERVs convertible to XRVs, but we could not find any evidence for endogenization such as, especially, inheritance by vertical transmission, a major feature of ERVs. Thus, we concluded the Korean RfRV-like viruses to be XRVs in this study and revised the related contents in the abstract, importance, results (line 89-101), and discussion (line 236-243, 264-276).

2. How do the authors classify the taxonomy of the bats? Please add this to the methodology section. Who identifies the species and which methods are used for species identification (either molecular or otherwise)?

Answer) Bat species was confirmed by Professor Chul-Un Chung, who is a bat ecology expert and author of a "A field guide to Korean bats" and has devoted in this field, through external features such as size, color, and ear shape, as well as anatomical features including skull structure, tooth shape, and wing structure (lines 283-285).

"Book 2020. A field guide to Korean bats by Chul- Un Chung"

3. Please describe in detail all measures to prevent contamination, e.g., whether there is a dedicated PCR room, which controls were used in the extraction and dPCR procedures, etc.

Answer) We strictly comply with basic rules to prevent contamination

1. Pre-PCR work, including DNA extraction and PCR preparation before amplification, is conducted in separate areas from post-PCR work, which involves amplification.
2. Positive and negative controls are included in every PCR assay.
3. Accurate volumes are aspirated and dispensed, and precautions are taken to avoid splashing when pipetting liquids.
4. Lab technicians always wear fresh gloves when working in a PCR area and change gloves frequently.
5. Aseptic cleaning is performed before and after every PCR procedure to maintain cleanliness and prevent contamination.

4. Please describe in detail all criteria used for selecting the public sequence database used for phylogenetic inference, e.g., all complete sequences deposited up to period X in the NCBI.

The sequences used in the retroviruses were referenced from the sequence utilized in the phylogenetic classification by Cui et al. Additional detailed information regarding this is provided in Supplementary Table 2.

5. About the cell tropism experiment, it's unclear how many replicates were utilized. From how I've interpreted this, a single culture was conducted for each cell and tissue type (considered a single biological replicate), while triplicates were carried out for the dPCR assay (considered a technical replicate). I believe that even with statistical evaluation for the dPCR triplicates, it may not be possible to assess the difference between tissues, considering that a greater deviation would likely be observed in culture replicates.

Answer) I apologize for the ambiguous description regarding the PCR work, which may have caused confusion. To clarify, the triplicates in our study were technical replicates using a DNA sample extracted from a single homogenized tissue sample (10mg per each tissue) or a single cell culture.

In the quantification of provirus using the dPCR assay, DNA samples extracted from the homogenized tissue samples were directly subjected to the dPCR assay to assess variations in proviral loads of different tissue types. Consequently, significant differences in proviral load were observed in different tissues of an individual.

To enhance clarity in the description of the PCR work, certain parts have been revised or added (lines 86-101, 290-291, 305-330).

6. Could you provide more details about the discussion on the cellular tropism experiment?

Answer) The title 'cell tropism test' may lead to misconceptions regarding the primary objective of our study. Our main aim was virus isolation to confirm the presence of XRVs, hence we have changed the title to 'Bat Retrovirus Isolation' (line 374).

Unfortunately, based solely on the results of this study, it is challenging to derive further meaningful interpretations beyond confirming the presence of a receptor for the isolated RfRV-like virus in RAW 264.7 cells. I sincerely apologize for any confusion this may have caused.

Nevertheless, I believe that the Y4b strain holds potential for studying the possibility of transmission to other animals, such as through challenge tests or using primary cell cultures derived from other animals. Additionally, although the likelihood is low, further research on pathogenesis or the development of therapeutics could be valuable in the event of potential infection in humans or other animals in the future.

Re: Spectrum04323-23R1 (Identification and Characterization of Recent Endogenous Retrovirus in *Rhinolophus ferrumequinum* Bats)

Dear Dr. HeeChun Chung:

The manuscript has been accepted, based on the most updated revised version contacted through the email tool (i.e., removing 'endogenous' from the title and the entire notions). Please make sure of this issue with the editorial office during the production stage.

Your manuscript has been accepted, and I am forwarding it to the ASM production staff for publication. Your paper will first be checked to make sure all elements meet the technical requirements. ASM staff will contact you if anything needs to be revised before copyediting and production can begin. Otherwise, you will be notified when your proofs are ready to be viewed.

Sincerely,
Takamasa Ueno
Editor
Microbiology Spectrum